behaviour, evolution

arousal, behavioural context, chimpanzee, human, valence, vocalization

**Author for correspondence:**
Roza G. Kamiloğlu
e-mail: r.g.kamiloglu@uva.nl

# Human listeners' perception of behavioural context and core affect dimensions in chimpanzee vocalizations

Roza G. Kamiloğlu[1], Katie E. Slocombe[2], Daniel B. M. Haun[3] and Disa A. Sauter[1]

[1]Department of Psychology, University of Amsterdam, REC G, Nieuwe Achtergracht 129B, 1001 NK, Amsterdam, The Netherlands
[2]Department of Psychology, University of York, York, UK
[3]Max Planck Institute for Evolutionary Anthropology, Leipzig, Germany

RGK, 0000-0002-1018-2595; KES, 0000-0002-7310-1887

Vocalizations linked to emotional states are partly conserved among phylogenetically related species. This continuity may allow humans to accurately infer affective information from vocalizations produced by chimpanzees. In two pre-registered experiments, we examine human listeners' ability to infer behavioural contexts (e.g. discovering food) and core affect dimensions (arousal and valence) from 155 vocalizations produced by 66 chimpanzees in 10 different positive and negative contexts at high, medium or low arousal levels. In experiment 1, listeners ($n = 310$), categorized the vocalizations in a forced-choice task with 10 response options, and rated arousal and valence. In experiment 2, participants ($n = 3120$) matched vocalizations to production contexts using yes/no response options. The results show that listeners were accurate at matching vocalizations of most contexts in addition to inferring arousal and valence. Judgments were more accurate for negative as compared to positive vocalizations. An acoustic analysis demonstrated that, listeners made use of *brightness* and *duration* cues, and relied on *noisiness* in making context judgements, and *pitch* to infer core affect dimensions. Overall, the results suggest that human listeners can infer affective information from chimpanzee vocalizations beyond core affect, indicating phylogenetic continuity in the mapping of vocalizations to behavioural contexts.

## 1. Introduction

When we hear a hissing cat or a person laughing, we may be able to infer information from these vocalizations, including both the individual's affective state and the kind of situation they are in. In 1872, Darwin [1] hypothesized that emotional vocal expressions have ancient evolutionary roots and that they are based on shared mechanisms across mammalian species. In the research on phylogenetic continuity of emotional vocalizations that have followed since then, researchers have primarily focused on vocal production; this work has established considerable similarities in the acoustic features linked to affective information in different animal groups. In an extensive review, Briefer [2] notes consistent acoustic correlates of core affect dimensions such as arousal (physiological alertness or attentiveness [3]) and valence (degree of positivity or negativity [3]) in vocalizations across mammalian species. Across species, there is thus consistency in the acoustic features that characterize arousal and valence.

In inferring affective information from vocalizations, perceivers might be able to make use of consistencies in affective vocalizations. When listening to conspecific or heterospecific vocalizations, accurate perception of the producer's affective state is beneficial for the perceiver in many contexts [4]. Indeed, inferring affective states from conspecific vocalizations can be essential for the perceiver in contexts including parental behaviour and sexual partner selection. Going beyond

conspecific vocalizations, listening to heterospecific vocalizations can be used to gather information about the producer's inner state which can facilitate adaptive behaviour in various contexts, like being attacked by a predator. This ability might be based on inherent capacities to perceive phylogenetically conserved acoustic regularities [4].

## (a) Human perception of affective information from heterospecific vocalizations

Most of what we know about human listeners' perception of affective information in heterospecific vocalizations comes from studies on core affect. This work has showed that humans can accurately infer arousal and valence from vocalizations of many different species [5–13]. However, acoustic features of mammalian vocalizations vary systematically across different types of behavioural contexts such as threats, food and play, that do not only vary in terms of arousal and valence [14,15]. Perceptual mechanisms may exist that allow human listeners to infer richer affective information from particular types of behaviours than inferences of core valence [16]. For instance, humans associate cats' purring with contentment and dogs' yelping with distress. However, it is not straightforward to map the affective information in heterospecific expressions onto human emotion categories, and there is a clear risk of anthropomorphizing those species. An alternative approach is to examine mappings between vocalizations and behavioural contexts as an indirect route to inferring affective states.

Only a few studies to date have tested human listeners' perception of behavioural contexts from heterospecific vocalizations. The results have shown that listeners can correctly classify the production context of dogs' barks [17], cats' meows [18] and the vocalizations of pigs [13]. However, previous studies are limited to domesticated animals that are distantly related to humans. Here, we seek to examine human listeners' ability to infer behavioural context and core affect dimensions from vocalizations of chimpanzees (*Pan troglodytes*), one of the genetically closest living relatives to humans.

In studies comparing animal vocalizations produced in positive and negative contexts, humans have consistently been found to be better at identifying affective information produced in negative contexts [8,13,18,19]. For instance, listeners correctly identified arousal levels in silver fox vocalizations only when they were produced in negative contexts [8]. In the current study, we, therefore, examine whether human listeners' perception of behavioural context and core affect dimensions is more accurate for negative as compared to positive chimpanzee vocalizations.

## (b) The present study

Drawing on two complementary approaches to phylogenetic continuity in emotional expressions, we sought to test the hypotheses that (i) human listeners can accurately infer the type of behavioural context in which chimpanzee vocalizations were produced [16]; and that (ii) human listeners can correctly judge arousal and valence from chimpanzee vocalizations [12]. We included chimpanzee vocalizations produced in a wide range of different positive and negative behavioural contexts at high, medium or low arousal levels.

In experiment 1, participants were asked to complete a forced-choice context categorization task for each vocalization, and to rate arousal and valence. We predicted that listeners

would be able to categorize the behavioural contexts and to judge the arousal level and valence from the vocalizations at better-than-chance levels. However, the 10-way forced-choice context categorization task was challenging for participants, and so experiment 2 employed a simpler paradigm. It tested whether participants could match the vocalizations to a corresponding behavioural context when selecting from two options (match versus no match). We predicted that listeners would be able to match vocalizations to their respective production contexts at better-than-chance levels. Finally, in both experiments, we expected that accuracy would be better for vocalizations produced in negative, as compared to positive, contexts.

In order to investigate the features shaping human listeners' perception of affective information from chimpanzee vocalizations, we conducted an acoustic analysis. First, we examined whether behavioural context, arousal level, and valence would be reflected in the acoustic structure of the vocalizations. Second, we tested which acoustic features would predict humans' perceptual judgements. The hypotheses, methods (including exclusion criteria), and data analysis plan for both experiments were pre-registered on the Open Science Framework (osf.io/mkde8) before data collection was commenced.

## 2. Experiment 1: categorization of behavioural contexts and judgements of arousal level and valence

In experiment 1, we tested whether human listeners would be able to accurately (i) categorize the behavioural context in which the chimpanzee vocalizations were produced by selecting from 10 context categories; and (ii) judge the arousal level (high, medium, low) and valence (positive, negative) of these vocalizations.

## (a) Participants

The sample size was predetermined by a power analysis using G*Power 3.1 [20] for a *t*-test given $d = 0.2$, power $= 0.80$, $\alpha = 0.005$. The power analysis was conducted based on the context categorization task, as we expected it to be the most difficult for participants. This categorization task included separate tests for 10 behavioural context categories; thus Bonferroni-corrected alpha level was used ($\alpha = 0.005$ [0.05/10]), and so 296 participants were required to detect a small effect size. To ensure that the study was not underpowered, data were collected from 14 additional participants to allow for potential exclusions (see Statistical analyses for exclusion criteria). Consequently, 310 participants (195 female, mean $(M)_{age} = 22.08$, s.d.$_{age} = 3.39$, range = 18–38 years old) took part in the experiment. All reported having no hearing impairments and no experience working with or studying chimpanzees. Participants were recruited via the University of Amsterdam, Department of Psychology's research pool, and flyers distributed across the university campus. The average duration of the main experiment was 27.43 min (s.d. = 9.75), and participation was compensated with monetary reward or course credit.

## (b) Materials and procedure
### (i) Stimuli
In the practice trials, two chimpanzee vocalizations taken from findsounds.com were used as stimuli. In the main task, the

**Table 1.** Behavioural contexts and core affect dimensions of chimpanzee vocalizations. (Note. For each context, vocalizations were obtained from between 4 and 21 individual chimpanzees.)

| | positive (*n* = 80) | negative (*n* = 75) | no specific valence (*n* = 11) |
|---|---|---|---|
| high arousal (*n* = 62) | pant hoots when discovering a large food source (*n* = 12) | Waa barks while threatening an aggressive chimp or predator (*n* = 16) | |
| | | victim screams when attacked by another chimpanzee (*n* = 21) | |
| | | alarm calls when discovering something scary (*n* = 13) | |
| medium arousal (*n* = 71) | rough grunts while eating high value food (*n* = 19) | tantrum screams when refused access to food (*n* = 15) | copulation calls while having sex (*n* = 11)[a] |
| | laughter while being tickled (*n* = 16) | whimpers by juveniles when separated from mother (*n* = 10) | |
| low arousal (*n* = 22) | rough grunts while eating low value food (*n* = 22) | | |

[a]Copulation calls may be associated with either positive (pleasure) or negative (fear/pain) valence, thus no specific valence is attributed to the vocalizations produced during copulation.

stimuli were 155 vocalizations produced by 66 individual chimpanzees in 10 types of behavioural contexts including positive and negative contexts at high, medium, and low arousal levels (table 1). The behavioural contexts were recorded by author K.E.S. in real time, alongside the sound recordings of vocalizations, and K.E.S., an expert in chimpanzee vocal communication, provided classifications of the arousal levels and valence of each call type (table 1). Descriptions of behavioural contexts and classification of each context based on arousal level and valence, together with number of stimuli, are listed in table 1; details of the recording set-ups are provided in the electronic supplementary material, Text S1. A representative vocalization for each context can be found in the electronic supplementary material, Audio S1. All recordings were normalized for peak amplitude using AUDACITY software (http://audacity.sourceforge.net) before the experiment.

### (ii) Experimental procedure

On arrival at the laboratory, each participant was led to a silent individual cubicle. After completing two practice trials, participants listened to the 155 chimpanzee vocalizations and for each were asked to (i) make a forced-choice context categorization, selecting from 10 categories, (ii) indicate the level of arousal on a 5-point scale (1 = very low, 5 = very high), and (iii) indicate the emotional valence on a 5-point scale (1 = very negative, 5 = very positive). Finally, participants reported their familiarity with each behavioural context (how familiar are you with the chimpanzees in the context of X from zoo settings or media?), and a representative vocalization from each context (how familiar are you with this chimpanzee vocalization from zoo settings or media?) on a 5-point scale (1 = not at all, 5 = extremely).

The presentation order of vocalizations, scales, and context categories were randomized separately for each participant. Participants could replay each vocalization as many times as needed to make their judgments. The stimuli were presented through headphones (Monacor MD-5000DR) connected to a computer, and the sound level was held constant across participants. The experimental interface was created using PSYCHOPY [21].

### (c) Statistical analyses

Before analysis, the dataset was checked for outliers, defined as performance of three s.d. or more below the mean on the categorization task. No participants had to be excluded.

To test whether human listeners would perform better than chance in the categorization of contexts, the proportion of correct responses was calculated for each participant for each context category. Unbiased hit rates (Hu scores, [22]) were calculated to control for individual biases in the use of particular context categories. These were arcsine transformed before the analysis to stabilize variance [22]. Following this transformation, all variables were checked for normality using a Shapiro–Wilk test, which indicated that they were not normally distributed ($p$s < 0.001). We therefore employed paired sample Wilcoxon signed-rank tests. Chance levels were calculated for each context per individual following Wagner's formula [22]: the product of the column and row marginals, divided by the squared number of observations. The corrected chance level takes the number of stimuli for each context category into account. Arcsine-transformed Hu scores were then compared to chance using a paired sample Wilcoxon signed-rank test for each category, Bonferroni corrected for multiple comparisons (0.05/10).

To assess how accurately human listeners judged the arousal level and valence of the vocalizations, ratings on the 5-point scales were transformed into −2 (very low), −1 (low), 0 (medium), 1 (high), 2 (very high); and −2 (very negative), −1 (negative), 0 (neutral), 1 (positive), 2 (very positive). A response was considered correct if (i) arousal ratings were significantly higher (lower) than zero for high (low) arousal vocalizations, (ii) arousal ratings were not significantly different from zero for medium arousal vocalizations, (iii) valence ratings were significantly higher (lower) than zero for emotionally positive (negative) vocalizations. Based on these criteria, we calculated arcsine-transformed Hu scores for statistical

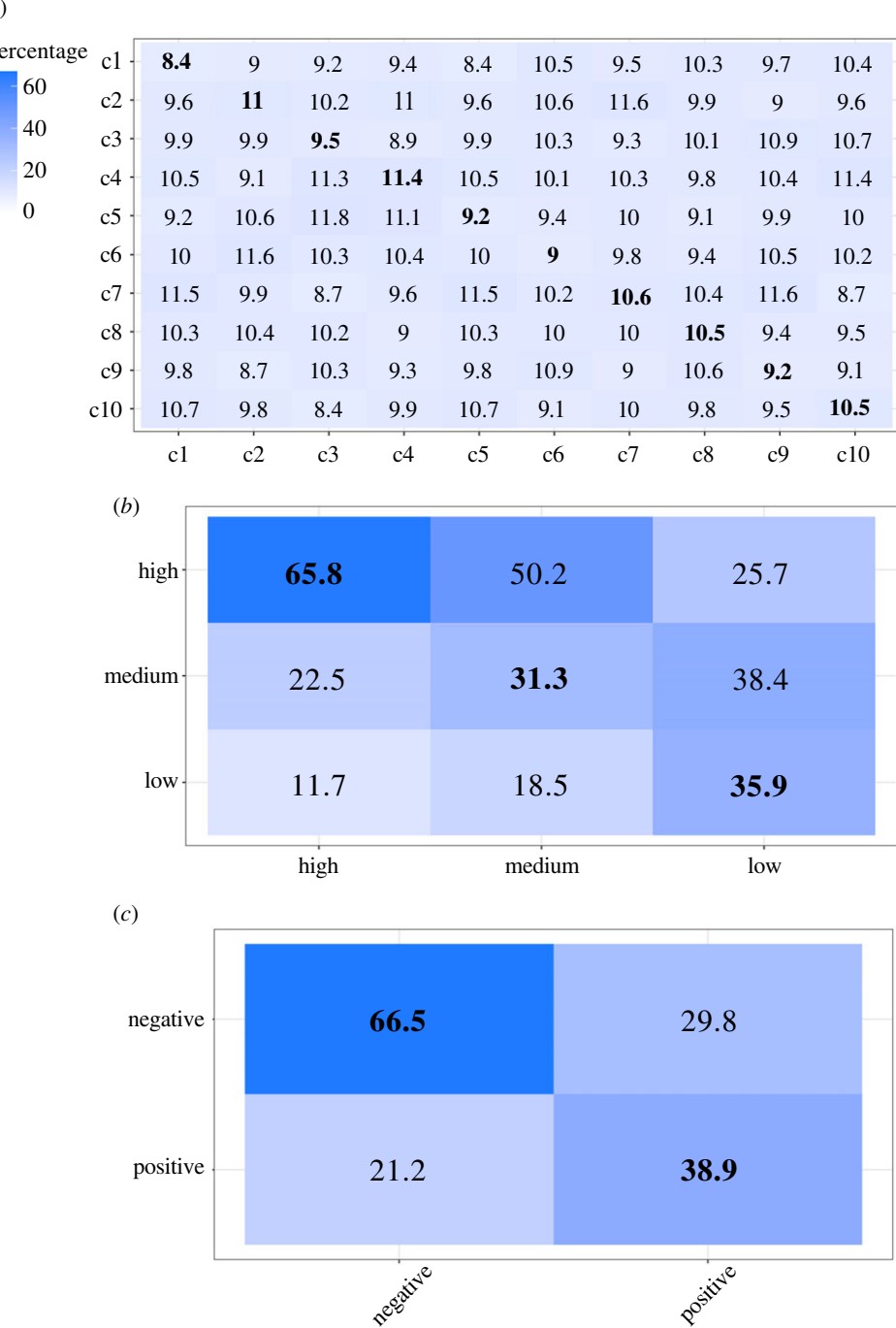

**Figure 1.** Heatmap of confusion matrices (%) for behavioural context categorization data (*a*), arousal (*b*) and valence (*c*) judgments. The *x*-axes represent stimulus types and the *y*-axes indicate responses. c1, eating high value food; c2, eating low value food; c3, copulating (having sex); c4, being separated from mother; c5, discovering a large food source; c6, being refused access to food; c7, being tickled; c8, being attacked by another chimpanzee; c9, threatening an aggressive chimpanzee; c10, discovering something scary. (Online version in colour.)

tests. Using separate Wilcoxon signed-rank tests, we tested whether performance was better than chance level for high, medium, and low arousal and for positive and negative vocalizations. Finally, we tested separately for context categorization, arousal and valence ratings whether perception accuracy was higher for vocalizations produced in negative as compared to positive contexts using paired sample Wilcoxon signed-rank tests with the arcsine-transformed Hu scores.

## (d) Results

Confusion matrices for average recognition percentages are shown in figure 1. The results show that participants were

not able to accurately categorize any of the behavioural contexts ($p$s > 0.005, Bonferroni corrected; figure 1*a*). However, judgments of core affect were significantly better than chance for high ($z = 14.734$, $p < 0.001$), low ($z = 13.567$, $p < 0.001$) and medium ($z = 8.745$, $p < 0.001$) arousal levels (figure 1*b*), as well as positive ($z = 14.805$, $p < 0.001$) and negative ($z = 14.713$, $p < 0.001$) valence (figure 1*c*).

In the analysis comparing listeners' performance for vocalizations produced in negative versus positive contexts, the judgements of behavioural contexts were not employed, as participants were unable to identify any of the contexts at better-than-chance levels. The results showed that, consistent with our prediction, participants were more accurate at identifying high arousal from vocalizations produced in

negative contexts than in positive contexts ($z = 14.374$, $p < 0.001$). However, listeners were more accurate at inferring medium arousal levels from positive as compared to negative vocalizations ($z = 14.852$, $p < 0.001$). Because the low arousal context category consisted of only one context, vocalizations from negative and positive contexts could not be compared. In terms of valence, listeners were better at judging the valence of vocalizations produced in negative as compared to positive contexts ($z = 15.146$, $p < 0.001$). To assess whether participants tend to perceive vocalizations as more negative or positive in general, we calculated the average of valence ratings across positive and negative vocalizations per participant ($M = -0.29$, s.d. $= 1.11$). When the valence ratings were compared against zero, they show a bias towards judging the vocalizations as negative ($z = -14.787$, $p < 0.001$). Human listeners tend to perceive chimpanzee vocalizations as more negative in general.

On average, participants rated both behavioural contexts ($M = 1.86$, s.d. $= 0.89$) and representative vocalizations ($M = 2.14$, s.d. $= 0.98$) as unfamiliar. Because of the large number of stimuli and judgements, we checked for evidence of fatigue by comparing the accuracy in early (the first 30) and late (the last 30) trials. Pairwise comparisons showed that participants' performance on the arousal judgement task was high in both the early ($M = 46.74$, s.d. $= 0.10$) and late trials ($M = 44.92$, s.d. $= 0.11$), although participants performed better in the early trials ($z = 2.552$, $p = 0.011$). No difference in accuracy was found for early and late judgments of context categorization and valence (see the electronic supplementary material, table 1S for details). It is therefore unlikely that participants' performance was adversely affected by possible fatigue.

# 3. Experiment 2: matching chimpanzee vocalizations to a single behavioural context

The 10-way context categorization task used in experiment 1 was challenging for participants because they were asked to choose from a substantial number of unfamiliar behavioural context categories. In experiment 2, we therefore sought to test whether listeners would be able to match vocalizations to behavioural contexts in a simpler task involving a single behavioural context for each participant.

## (a) Participants

Each participant was given a context matching task for a single context with yes/no response options on each trial. A power analysis (G*Power 3.1; [20]) based on a $t$-test given $d = 0.2$, power $= 0.80$, $\alpha = 0.05$ showed that 156 judgments per stimulus were needed. To reduce the risk of learning effects, each participant heard half of the stimuli from the target behavioural context category. This yielded a total number of 312 participants per context category. Because we tested 10 behavioural contexts, the total sample size was set to 3120. Consequently, a total of 3120 participants (1570 females,[1] $M_{age} = 34.03$,[2] s.d.$_{age} = 10.34$, range $= 18–75$ years old) were recruited from Amazon Mechanical Turk to take part in the experiment. All reported having no hearing impairments or experience of working with or studying chimpanzees. Each session lasted around 10 min and participation in the experiment was compensated with monetary reward.

## (b) Materials and procedure
### (i) Stimuli
Experiment 2 used the same stimuli as those in experiment 1 for both the practice trials and the main task.

### (ii) Experimental procedure
The study was run online using the Qualtrics survey tool (Qualtrics, Provo, UT). Before commencing, participants were instructed to complete the experiment in a silent environment and use headphones. Participants were given two screening questions. On one they were played a doorbell and on the other a car horn sound. They were asked to indicate what they heard, with 'doorbell' and 'car horn' as response options. Participants who failed one or more screening questions were not able to continue to the main experiment.

After the practice and screening trials, each participant was randomly assigned to one of the 10 conditions, each focusing on a specific behavioural context. In each condition, participants were asked to give a match-to-context judgment (does this vocalization match context X?), selecting from yes and no options. The matching vocalizations were a randomly selected subset (half) of the vocalizations from that behavioural context. This constituted one-fourth of the stimuli heard by that participant; the other three-fourths were the non-matching stimuli randomly drawn from all of the other context categories. Only a quarter of the stimuli heard by a given participant were thus from the relevant behavioural context, again to reduce the risk of learning effects. The presentation order of vocalizations was randomized for each subject.

## (c) Statistical analyses
The dataset was checked for participants whose performance was three s.d. or more below the context-specific mean, but none were and so all data were retained.

We quantified participants' ability to match behavioural contexts using the sensitivity index $d$-prime. $d$-prime controls for individual biases in the use of a particular response, and is calculated as $z$-transformed hit rates minus false alarm rates [23]. Hit and false alarm rates with extreme values (i.e. 0 or 1) return an error when $z$-transformed. Those cases are commonly adjusted by replacing rates of zero with $0.5/n$ ($0.5/m$) and rates of 1 with $(n-0.5)/n$ ($[m-0.5]/m$) where $n$ ($m$) is the number of signal (noise) trials [24]. We calculated hit rates as the proportion of 'yes' trials to which participants responded yes, false alarm rates as the proportion of 'no' trials responded to as yes. In order to test our hypothesis that human listeners would perform better than chance in matching vocalizations to context types, $d$-prime scores for each participant were tested against chance (random guessing, reflected by a $d$-prime score of zero) using separate one sample $t$-tests for each context type at the Bonferroni-corrected level $\alpha$ level ($\alpha = 0.005$).

Furthermore, we tested whether performance would be better for negative than for positive vocalizations. This was tested using an ANOVA comparing the mean accuracy from negative versus positive behavioural contexts using $d$-prime scores.

## (d) Results
Mean accuracy levels ($d$-primes) per behavioural context are shown in figure 2. The statistical tests showed that participants were able to accurately match most of the vocalizations

**Figure 2.** $d$-prime scores per behavioural context showing human listeners' performance in matching vocalizations to production contexts. Bold indicates better than the chance level performance. (Online version in colour.)

to behavioural contexts. Specifically, performance was significantly better than chance for eating high value food ($t = 5.04$, $p < 0.001$, $d = 0.28$, 95% confidence intervals (CIs) (0.15, 0.34)), eating low value food ($t = 9.59$, $p < 0.001$, $d = 0.55$, 95% CIs (0.49, 0.75)), discovering a large food source ($t = 5.52$, $p < 0.001$, $d = 0.31$, 95% CIs (0.17, 0.37)), being refused access to food ($t = 13.09$, $p < 0.001$, $d = 0.74$, 95% CIs (0.63, 0.85)), being attacked by another chimpanzee ($t = 22.99$, $p < 0.001$, $d = 1.29$, 95% CIs (1.06, 1.26)), and threatening an aggressive chimp or predator ($t = 11.19$, $p < 0.001$, $d = 0.64$, 95% CIs (0.37, 0.53)). Performance was not better than chance, however, for vocalizations of copulation (having sex) ($t = -2.99$, $p = 0.003$, $d = 0.17$, 95% CIs ($-0.04$, 0.003)), being separated from mother ($t = 2.81$, $p = 0.005$, $d = 0.16$, 95% CIs (0.04, 0.23)), being tickled ($t = 1.77$, $p = 0.19$, $d = 0.10$, 95% CIs ($-0.01$, 0.19)), and discovering something scary ($t = -16.49$, $p = 0.003$, $d = 0.93$, 95% CIs ($-0.68$, $-0.53$)). Accuracy levels for matching vocalizations produced in negative contexts were significantly better than those from positive contexts (negative: $M = 0.43$, s.d. $= 0.37$, positive: $M = 0.22$, s.d. $= 0.47$, $F_{1,631} = 38.938$, $p < 0.001$).

## 4. Acoustic analysis

We performed an acoustic analysis to explore acoustic features shaping human perception of affective information in chimpanzee vocalizations. First, independently of perceptual responses of listeners, a classification analysis was conducted to test whether chimpanzee vocalizations differ by context, arousal level, and valence, in terms of acoustic features. We then examined which acoustic features, if any, would predict humans' ability to accurately infer affective information from chimpanzee vocalizations in terms of correctly judging the arousal level and valence of the vocalizations (experiment 1) and accurately matching the vocalizations to the corresponding behavioural context (experiment 2).

### (a) Method
#### (i) Extraction of acoustic features from chimpanzee vocalizations
We measured acoustic features of 155 vocalizations produced by 66 individual chimpanzees using PRAAT [25]. For each vocalization, we measured the following acoustic features: *number of calls in a bout, duration of each call, time of the maximum* *peak frequency, relative position of the peak frequency within a call, percentage of voiced frames, jitter, shimmer, spectral centre of gravity (SCoG)* as well as *minimum, maximum, mean* and *s.d. of fundamental frequency ($f_0$)* and *harmonics-to-noise ratio (HNR)*. We based the choice of parameters on previous findings on production and perception of affective mammalian vocalizations: *duration, $f_0$* and *HNR* are linked to the affective state of the caller across many animal species [4,14,26,27]. *Peak frequency* has been found to differ across dog barks recorded in different contexts [17]. In addition, the *percentage of voiced frames* was added as a tonality measure because in nonverbal human vocalizations such as laughter, voiced frames are typically more periodic, while unvoiced frames are noisier and more aperiodic [28]. *Jitter* and *shimmer* are important parameters for analysis of arousal in animal vocalizations [29], while *SCoG* is associated with the perception of arousal in humans [6,7,30]. Means and s.d. of all acoustic parameters can be found in the electronic supplementary material, table 2S.

### (ii) Further selection of acoustic features
To avoid multicollinearity, we performed a principal component analysis (PCA) with varimax rotation on the 15 acoustic parameters to attempt to reduce the number of acoustic parameters. Based on the examination of the scree plot and selecting components that explain more than 10% of the variance, the first three components, together explaining 63% of the variance, were retained. Factor loadings on the three acoustic dimensions can be found in the electronic supplementary material, table 3S. Online interactive maps showing the distribution of the 10 behavioural contexts is available in https://emotionwaves.github.io/context/, arousal levels in https://emotionwaves.github.io/arousal/, and valence in https://emotionwaves.github.io/valence/ on the first three acoustic dimensions. These visualizations demonstrate that the behavioural contexts, arousal levels, and valence are reflected in the acoustic structure of vocalizations. The first dimension mainly relates to *HNR*, which is a measure of clear versus noisy components in the signal. The second dimension is primarily related to *pitch*, while the third mainly relates to temporal measures. The variance of inflation factor (VIF) was substantially greater than 1 for acoustic features loading on the second and third dimensions (*duration*: 4.62; $f_0$ *min*: 30.42; $f_0$ *max*: 64.16; $f_0$ *mean*: 37.86; $f_0$ *s.d.*: 21.05;

*time of the maximum peak frequency*: 4.10) indicating that there was a collinearity problem [31]. We therefore selected the features with highest interpretability based on the previous literature in addition to the factor loadings on the first three components. This selection allowed us to choose acoustic features with low VIF and high factor loadings on the first three dimensions. The selected acoustic features for statistical analyses were thus: *SCoG*, *duration*, $f_0$ *mean*, $f_0$ *s.d.*, *HNR mean*, and *HNR max*. Collinearity was not a problem for these features (VIF: *SCoG*: 1.67; *duration*: 1.47; $f_0$ *mean*: 1.62; $f_0$ *s.d.*: 1.25; *HNR mean*: 1.56; *HNR max*: 1.62).

## (b) Statistical analyses

We sought to test whether behavioural contexts, arousal levels and valence could be differentiated based on the selected acoustic features. Multinomial logistic regressions (MLR) were performed in SPSS (Version 23, IBM Statistics) on the acoustic features to determine whether the acoustic parameters provide sufficient information to predict the actual behavioural contexts and arousal levels, and binomial logistic regression (BLR) for valence.

To assess which, if any, acoustic parameters of the vocalizations would map onto listeners' ability to accurately perceive (i) behavioural context, (ii) arousal levels, and (iii) valence, we conducted three generalized linear mixed models (GLMMs). The dependent variable was a binary response (i.e. correct or incorrect response). Participant and chimpanzee identities were entered as random factors, accounting for participant and chimpanzee variability. The selected acoustic parameters were set as fixed factors. We used Akaike's information criterion (AIC) to select the most parsimonious model [32]. $\Delta$AICs are calculated as the difference between the AICc of the fitting model and the best model to identify the models with the highest power to explain the variation in the dependent variable. Lower AIC values indicate improved support for each model [32,33], and each added variable is considered to improve the fit only if it increases the AIC value by more than two units [34]. GLMMs were implemented using lme4 package [35] with optimizer 'bobyqa' [36]. Binomial data and estimated odds were plotted as forest plots for fixed effects 'sjplot' package in R [37].

## (c) Results

### (i) Classification of behavioural contexts, arousal levels and valence based on acoustic parameters

MLR on behavioural contexts showed that the overall model was significant $\chi^2_{54} = 595.618$, $p < 0.001$. All acoustic parameters, *SCoG* ($\chi^2_9 = 92.919$, $p < 0.001$), *duration* ($\chi^2_9 = 114.154$, $p < 0.001$), $f_0$ *mean* ($\chi^2_9 = 92.283$, $p < 0.001$), $f_0$ *s.d.* ($\chi^2_9 = 50.906$, $p < 0.001$), *HNR mean* ($\chi^2_9 = 112.324$), $p < 0.001$ and *HNR max* ($\chi^2_9 = 22.620$, $p < 0.01$) made significant unique contributions and the overall model showed 85.7% classification agreement on behavioural context classification.

The results from the MLR on arousal levels revealed that the overall model was significant ($\chi^2_2 = 191.391$, $p < 0.001$). Significant contributions were made by *SCoG* ($\chi^2_2 = 28.990$, $p < 0.001$), *duration* ($\chi^2_2 = 72.489$, $p < 0.001$) and *HNR mean* ($\chi^2_2 = 25.352$, $p < 0.001$). Vocalizations with higher arousal levels were longer in *duration* compared to vocalizations with lower arousal levels. *HNR mean* was higher for high and medium arousal and lower for low arousal vocalizations,

while the *SCoG* of low arousal vocalizations was lower than that of medium and high arousal vocalizations. The final model showed a classification agreement of 83.1%.

Third, the BLR on valence showed that the overall model was significant ($\chi^2_6 = 60,433$, $p < 0.001$). *Duration* ($\chi^2_6 = 8.789$, $p < 0.01$), $f_0$ *mean* ($\chi^2_6 = 19.797$, $p < 0.01$) and *HNR mean* ($\chi^2_6 = 5.381$, $p < 0.05$) made significant unique contributions. *Duration* was longer for negative vocalizations and $f_0$ *mean* and *HNR mean* were higher for negative vocalizations than positive vocalizations. The final model had a classification agreement of 73.4%.

### (ii) Prediction of human listeners' perceptual judgments from acoustic parameters

GLMMs revealed that *SCoG* ($z = 6.59$, $p < 0.001$), *duration* ($z = 2.83$, $p < 0.01$), $f_0$ *s.d.* ($z = -2.73$, $p < 0.01$), *HNR mean* ($z = -6.03$, $p < 0.001$) and *HNR max* ($z = 3.31$, $p < 0.001$) significantly predicted accurate match-to-context responses in experiment 2. *SCoG* is a measure of how high the frequencies in a spectrum are, which is perceptually connected with the impression of *brightness* of a vocalization. *Duration* refers to the total duration of calls in whole stimulus, while $f_0$ is the lowest periodic cycle of the acoustic signal, which has the perceptual correlate of *pitch*. *HNR* is the degree of acoustic periodicity, which relates to human perception of *noisiness*. The model selection procedure based on the AIC identified the model excluding $f_0$ *mean* as the strongest model for explaining variation in human listeners' accurate responses in the match-to-context task. The best predictor of performance was *SCoG*, which was linked to participants' ability to correctly match vocalizations to behavioural contexts (figure 3).

GLMM predicting accurate arousal level judgments in experiment 1 revealed significant effects of *SCoG* ($z = 5.33$, $p < 0.001$), *duration* ($z = 2.91$, $p < 0.05$), $f_0$ *mean* ($z = 13.25$, $p < 0.001$) and $f_0$ *s.d.* ($z = 13.90$, $p < 0.001$). Increases in those acoustic parameters predicted higher accuracy in listeners' judgments of arousal level. The best predictor of arousal level judgments was $f_0$ *s.d.* Specifically, decreases in this parameter (corresponding approximately to less pitch variability) predicted better listener accuracy in identification of arousal levels. For valence judgments, *SCoG* ($z = 11.96$, $p < 0.001$), *duration* ($z = 8.24$, $p < 0.001$), $f_0$ *mean* ($z = 15.48$, $p < 0.001$) and $f_0$ *s.d.* ($z = -5.78$, $p < 0.001$), showed significant effects on the prediction of listeners' performance. Specifically, increases in *SCoG*, *duration* and $f_0$ *mean* predicted more accurate valence judgments, while increases in $f_0$ *s.d.* predicted lower accuracy. The best predictor of valence judgements was $f_0$ *mean*, which predicted better accuracy in the identification of valence. In explaining variation in human listeners' accuracy in identifying both arousal levels and valence from chimpanzee vocalizations, the model excluding *HNR mean* as well as the model without *HNR max* were the strongest models. The effects of each acoustic features on the accurate perception of behavioural context, arousal levels and valence are visualized in figure 3. Full details of the GLMMs and model selection procedures are provided in the electronic supplementary material, tables 4S and 5S.

## 5. Discussion

Two experiments tested human listeners' ability to accurately (i) perceive behavioural contexts in which chimpanzee vocalizations were produced, using a 10-way context categorization

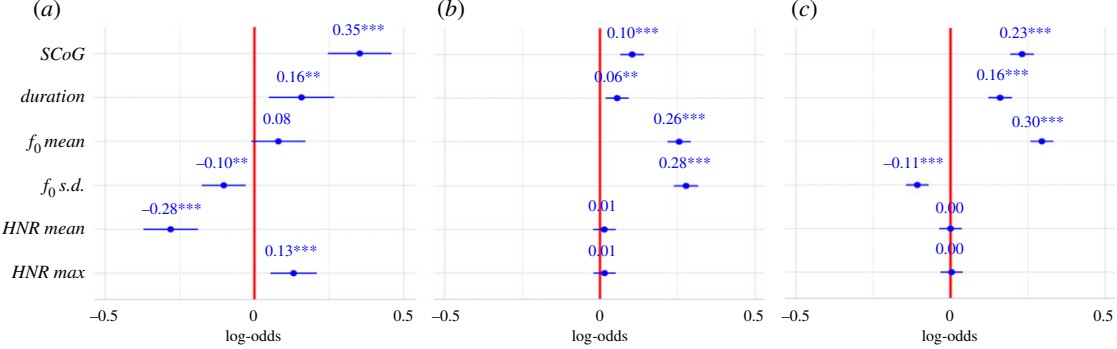

**Figure 3.** Forest plots of estimates of the GLMMs. Estimates for fixed effects are given as log-odds. The vertical intercept indicates no effect. (a) Behavioural context based on match-to-context task in experiment 2, (b) arousal level judgment task in experiment 1, (c) valence judgement task in experiment 1. (Online version in colour.)

task and a yes/no match-to-context task; and judge (ii) arousal and valence from chimpanzee vocalizations. Human listeners failed to categorize production contexts of vocalizations when a 10-way forced-choice task was used. However, they were able to match vocalizations to most behavioural contexts in the simpler yes/no match-to-context task. In addition, the arousal levels (high, medium, low) and valence (positive, negative) of the chimpanzee vocalizations were accurately inferred by human listeners. Overall, participants performed better with negative, as compared to positive, vocalizations.

In experiment 1, participants were asked to select the best matching context from 10 unfamiliar behavioural context categories. Such tasks are difficult for listeners as it is more challenging to evaluate and compare contexts [4]. Moreover, 10 is a large number of options for a categorization task. It has been suggested that even though increasing the number of alternatives in forced-choice tasks has advantages (e.g. reducing the guessing rate), for a given task, there is a point at which the number of options becomes too large for participants [38]. The present results suggest that for human listeners to be able to accurately map chimpanzee vocalizations to 10 unfamiliar behavioural contexts, participants may require additional information about the contexts, and/or information carried by other channels such as facial expressions.

In experiment 2, when a yes/no match-to-context task was used, listeners accurately matched the vocalizations produced while eating high and low value food, discovering a large food source, being refused access to food, being attacked by another chimpanzee, and threatening an aggressive chimpanzee or predator. Given that listeners in our experiment had minimal prior exposure to chimpanzees, they are unlikely to have learned to decode chimpanzee vocalizations. Rather, accurately mapping heterospecific vocalizations to behavioural contexts linked to affective states may draw on acoustic regularities that are conserved across related species. For instance, African elephants can differentiate between threatening and non-threatening human vocalizations [39], and Japanese sika deer uses the vocalizations produced by Japanese macaques when they discover a food source to locate fruit [40]. In these contexts, understanding heterospecific vocalizations clearly benefits the perceiver, and thus may confer a fitness advantage. To assess the effect of different degrees of acoustic regularities in vocalizations on perception of behavioural contexts from heterospecific vocalizations, future studies should aim at including vocalizations from multiple species differing in phylogenetic closeness.

Listeners failed to match vocalizations of copulation, being separated from mother, being tickled and discovering something scary. A possible explanation is that there may be a great deal of variability in the vocalizations produced in these contexts, depending on factors such as who potential perceivers are (e.g. kin versus non-kin, allies versus competitors). For instance, female chimpanzee copulation calls have been found to differ when copulating with high ranking males compared to low ranking males [41]. Thus, listeners might need additional contextual information to be able to specify vocalizations produced in certain type of contexts, or might not be able to identify certain contexts from vocalizations at all.

In general, listeners' judgments of negative behavioural contexts were more accurate than judgments of positive contexts. Similarly, high arousal vocalizations and valence were more accurately inferred from vocalizations produced in negative contexts. In particular, accuracy was especially high for highly aroused negative vocalizations, which might signal immediate, potentially dangerous situations. It has been proposed that stronger phylogenetic continuity for negative affective signals may be a result of a homologous signalling system that benefits species in dangerous contexts [7]. From this perspective, the acoustic structure of vocalizations produced in negative contexts may be more likely to have been conserved, because negative contexts involve risks. Survival might be facilitated by the ability to recognize vocalizations produced in negative contexts not only by conspecifics, but also by members of other species [42]. Indeed, cross-species 'eavesdropping' on alarm calls has been suggested to increase chances of survival [43]. Thus, acoustic structure may have been preserved to a greater degree for negative as compared to positive vocalizations.

Independently of listeners' perceptual responses, acoustic features of chimpanzee vocalizations varied systematically across different behavioural contexts, arousal levels, and valence. Listeners used *brightness*, *duration*, *pitch variation*, *noisiness* and *maximum level of noisiness* to make accurate classifications of vocalizations into behavioural contexts. *Brightness*, *duration*, *pitch* and *pitch variability* predicted listeners' ability to correctly infer both arousal levels and valence. *Noisiness* of vocalizations was a more useful acoustic feature in matching production contexts compared to other features, while more simple acoustic features like *pitch mean* and *pitch variation* were more effective in identification of arousal and valence. In line with our findings, Maruščáková and colleagues [11]

found that simple acoustic features such as *pitch* were more useful in human judgments of valence than *noisiness* in piglet vocalizations. Similarly, Filippi and colleagues [7] have shown that humans mainly rely on *pitch* to identify high arousal vocalizations across nine vertebrate species. Furthermore, consistently with our findings, *duration* and *brightness* have also been suggested to be effective acoustic features in humans' ability to identify arousal level from vocalizations [5,7,30]. In summary, acoustic analysis revealed that chimpanzee vocalizations differ by context, arousal and valence based on acoustic features and allowed us to identify specific features contributing to human listeners' perceptual judgments.

In conclusion, the present study demonstrates that human listeners can accurately perceive affective information beyond core affect dimensions from the vocalizations of a closely related species, chimpanzees. These findings suggest phylogenetic preservation of acoustic features mapping onto specific behavioural contexts, as well as features characterizing arousal levels and valence.

Ethics. The School of Psychology Ethics Committee, University of St Andrews gave ethical clearance for the non-invasive, behavioural studies that included the recording of chimpanzee vocalizations. Experiment 1 (project no. 2018-SP-9517) and experiment 2 (project no. 2019-SP-10653) were approved by the Ethics Review Board of the Faculty of Social and Behavioural Sciences, University of Amsterdam. All participants provided informed consent before participation.

Data accessibility. Data are available from https://figshare.com/s/83157a7974c4659009f1 and reproducible scripts for data analysis are available from https://figshare.com/s/eb570389986c95331907.

Competing interests. We declare we have no competing interests.

Funding. R.G.K. and D.A.S. are supported by ERC Starting grant no. 714977 awarded to D.A.S.

Acknowledgements. We would like to thank Thibaud Gruber for his helpful suggestions for the second experiment.

## Endnotes

[1] Eight participants preferred not to indicate their gender.
[2] Three participants are excluded in the descriptive statistics on age because of birth date errors.

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
