## [Reviewer comments · Proceedings of the Royal Society B: Biological Sciences]

Review History

RSPB-2020-0465.R0 (Original submission)

Review form: Reviewer 1

Recommendation

Accept with minor revision (please list in comments)

Scientific importance: Is the manuscript an original and important contribution to its field?

Good

General interest: Is the paper of sufficient general interest?

Good

Quality of the paper: Is the overall quality of the paper suitable?

Good

Is the length of the paper justified?

Yes

Should the paper be seen by a specialist statistical reviewer?

No

Do you have any concerns about statistical analyses in this paper? If so, please specify them explicitly in your report.

No

It is a condition of publication that authors make their supporting data, code and materials available - either as supplementary material or hosted in an external repository. Please rate, if applicable, the supporting data on the following criteria.

Is it accessible?

Yes

Is it clear?

Yes

Is it adequate?

Yes

Do you have any ethical concerns with this paper?

No

Comments to the Author

This is an interesting study that looked at humans' ability to categorize level of arousal, valence, and behavioural context from nonhuman primate vocalizations. Participants were good at the arousal and valence categorizations, but poorer at context categorizations.

I have no concerns, but I do wonder whether the authors administered any questionnaires to participants to determine their level of experience with nonhuman animals' vocalizations.

Review form: Reviewer 2

Recommendation

Major revision is needed (please make suggestions in comments)

Scientific importance: Is the manuscript an original and important contribution to its field?

Good

General interest: Is the paper of sufficient general interest?

Excellent

Quality of the paper: Is the overall quality of the paper suitable?

Good

Is the length of the paper justified?

Yes

Should the paper be seen by a specialist statistical reviewer?

Yes

Do you have any concerns about statistical analyses in this paper? If so, please specify them explicitly in your report.

Yes

It is a condition of publication that authors make their supporting data, code and materials available - either as supplementary material or hosted in an external repository. Please rate, if applicable, the supporting data on the following criteria.

Is it accessible?

Yes

Is it clear?

Yes

Is it adequate?

No

Do you have any ethical concerns with this paper?

No

Comments to the Author

Human Listeners' Perception of Behavioural Context and Core Affect Dimensions in Chimpanzee Vocalisations
Kamiloğlu et al.

Overall, I found this manuscript to be a generally thorough and interesting examination of cross-species vocal perception. It contains very detailed acoustic analyses and also two main experiments to determine human perception of chimp calls, based on behavioural context in which the calls were uttered, as well as arousal and valence (two key components linked to emotions). The results mostly appear robust and convincing, except see my comments on Exp 1 below.

I suggest the following changes:

Please insert line numbers in order to make the lives of reviewers and editors a little easier.

Key words should be in alphabetical order.

The order of results should be changed. A full understanding of the Experiments 1 and 2 is based on section 3 of the results - Acoustic Analysis. Reading Experiment 1 at first, I thought the classification of arousal in calls was based on subjective assessments of one of the authors (See: Text S1: Recording of chimpanzee vocalisations)

Table 1. I assume the classification of calls according to arousal and valence is based on the results of the Acoustic Analysis section? However, this is not clear. For example, it is possible that, at least subjectively and without acoustic data, Tantrum screams might be considered High instead of Medium arousal, and Whimpers might be considered Low arousal instead of Medium.

Please provide examples of the calls types as Supplemental files. These should be available at the review stage.

Please insert the sample size for chimps used in the study and not just the number of vocalisations ($n = 155$). The chimp sample size used for sourcing the call examples also needs to be more prominent in both the Methods (e.g. page 7/32) and Results of the manuscript - currently it is quite difficult to locate.

Page 8/32. Experiment 1. "Participants listened to the 155 chimpanzee vocalisations". I have reservations about how informative this sort of setup could be (with 10 behavioural contexts, three arousal levels). How long (average and SD) did it take human participants to work their way through 155 chimp calls? Would you expect the same level of accuracy and focus in the

human subjects at call 10 or 14, versus call 145 or 150? There is no analysis reported that checks whether the human subjects were better at classifying the first 30 calls versus the last 30, for example.

Page 19/32. MNR?

The formatting of references contains many inconsistencies, which are not in the journal style.

Decision letter (RSPB-2020-0465.R0)

08-Apr-2020

Dear Ms Kamiloglu:

I am writing to inform you that your manuscript RSPB-2020-0465 entitled "Human Listeners' Perception of Behavioural Context and Core Affect Dimensions in Chimpanzee Vocalisations" has, in its current form, been rejected for publication in Proceedings B.

This action has been taken on the advice of referees, who have recommended that substantial revisions are necessary. With this in mind we would be happy to consider a resubmission, provided the comments of the referees are fully addressed. However please note that this is not a provisional acceptance.

Sincerely,
Dr Robert Barton
<mailto:proceedingsb@royalsociety.org>

Associate Editor

Board Member: 1

Comments to Author:

The two reviewers agree that the study is interesting and informative. There are, however, a number of areas where further clarification of the methodology is required. Reviewer 2 also raises important concerns about potential changes in the accuracy of classifications over time. It would be relatively straightforward to include additional analyses to address this issue.

Reviewer(s)' Comments to Author:

Referee: 1

Comments to the Author(s)

This is an interesting study that looked at humans' ability to categorize level of arousal, valence, and behavioural context from nonhuman primate vocalizations. Participants were good at the arousal and valence categorizations, but poorer at context categorizations.

I have no concerns, but I do wonder whether the authors administered any questionnaires to participants to determine their level of experience with nonhuman animals' vocalizations.

Referee: 2

Comments to the Author(s)

Human Listeners' Perception of Behavioural Context and Core Affect Dimensions in Chimpanzee Vocalisations
Kamiloğlu et al.

Overall, I found this manuscript to be a generally thorough and interesting examination of cross-species vocal perception. It contains very detailed acoustic analyses and also two main experiments to determine human perception of chimp calls, based on behavioural context in which the calls were uttered, as well as arousal and valence (two key components linked to emotions). The results mostly appear robust and convincing, except see my comments on Exp 1 below.

I suggest the following changes:

Please insert line numbers in order to make the lives of reviewers and editors a little easier.

Key words should be in alphabetical order.

The order of results should be changed. A full understanding of the Experiments 1 and 2 is based on section 3 of the results - Acoustic Analysis. Reading Experiment 1 at first, I thought the classification of arousal in calls was based on subjective assessments of one of the authors (See: Text S1: Recording of chimpanzee vocalisations)

Table 1. I assume the classification of calls according to arousal and valence is based on the results of the Acoustic Analysis section? However, this is not clear. For example, it is possible that, at least subjectively and without acoustic data, Tantrum screams might be considered High instead of Medium arousal, and Whimpers might be considered Low arousal instead of Medium.

Please provide examples of the calls types as Supplemental files. These should be available at the review stage.

Please insert the sample size for chimps used in the study and not just the number of vocalisations (n = 155). The chimp sample size used for sourcing the call examples also needs to

be more prominent in both the Methods (e.g. page 7/32) and Results of the manuscript – currently it is quite difficult to locate.

Page 8/32. Experiment 1. “Participants listened to the 155 chimpanzee vocalisations”. I have reservations about how informative this sort of setup could be (with 10 behavioural contexts, three arousal levels). How long (average and SD) did it take human participants to work their way through 155 chimp calls? Would you expect the same level of accuracy and focus in the human subjects at call 10 or 14, versus call 145 or 150? There is no analysis reported that checks whether the human subjects were better at classifying the first 30 calls versus the last 30, for example.

Page 19/32. MNR?

The formatting of references contains many inconsistencies, which are not in the journal style.

Author's Response to Decision Letter for (RSPB-2020-1148.R0)

See Appendix A.

RSPB-2020-1148.R1 (Revision)

Review form: Reviewer 2

Recommendation

Accept as is

Scientific importance: Is the manuscript an original and important contribution to its field?

Excellent

General interest: Is the paper of sufficient general interest?

Excellent

Quality of the paper: Is the overall quality of the paper suitable?

Excellent

Is the length of the paper justified?

Yes

Should the paper be seen by a specialist statistical reviewer?

No

Do you have any concerns about statistical analyses in this paper? If so, please specify them explicitly in your report.

No

It is a condition of publication that authors make their supporting data, code and materials available - either as supplementary material or hosted in an external repository. Please rate, if applicable, the supporting data on the following criteria.

Is it accessible?

Yes

Is it clear?

Yes

Is it adequate?

Yes

Do you have any ethical concerns with this paper?

No

Comments to the Author

n/a

Decision letter (RSPB-2020-1148.R0)

20-May-2020

Dear Ms Kamiloglu

I am pleased to inform you that your Review manuscript RSPB-2020-1148 entitled "Human Listeners' Perception of Behavioural Context and Core Affect Dimensions in Chimpanzee Vocalisations" has been accepted for publication in Proceedings B.

The referee(s) do not recommend any further changes. Therefore, please proof-read your manuscript carefully and upload your final files for publication. Because the schedule for publication is very tight, it is a condition of publication that you submit the revised version of your manuscript within 7 days. If you do not think you will be able to meet this date please let me know immediately.

To upload your manuscript, log into <http://mc.manuscriptcentral.com/prsb> and enter your Author Centre, where you will find your manuscript title listed under "Manuscripts with Decisions." Under "Actions," click on "Create a Revision." Your manuscript number has been appended to denote a revision.

You will be unable to make your revisions on the originally submitted version of the manuscript. Instead, upload a new version through your Author Centre.

- 1) A text file of the manuscript (doc, txt, rtf or tex), including the references, tables (including captions) and figure captions. Please remove any tracked changes from the text before submission. PDF files are not an accepted format for the "Main Document".
- 2) A separate electronic file of each figure (tiff, EPS or print-quality PDF preferred). The format should be produced directly from original creation package, or original software format. Please note that PowerPoint files are not accepted.

3) Electronic supplementary material: this should be contained in a separate file from the main text and the file name should contain the author's name and journal name, e.g. `authorname_procb_ESM_figures.pdf`

All supplementary materials accompanying an accepted article will be treated as in their final form. They will be published alongside the paper on the journal website and posted on the online figshare repository. Files on figshare will be made available approximately one week before the accompanying article so that the supplementary material can be attributed a unique DOI. Please see: <https://royalsociety.org/journals/authors/author-guidelines/>

4) Data-Sharing and data citation

It is a condition of publication that data supporting your paper are made available. Data should be made available either in the electronic supplementary material or through an appropriate repository. Details of how to access data should be included in your paper. Please see <https://royalsociety.org/journals/ethics-policies/data-sharing-mining/> for more details.

If you wish to submit your data to Dryad (<http://datadryad.org/>) and have not already done so you can submit your data via this link <http://datadryad.org/submit?journalID=RSPB&manu=RSPB-2020-1148> which will take you to your unique entry in the Dryad repository.

Once again, thank you for submitting your manuscript to Proceedings B and I look forward to receiving your final version. If you have any questions at all, please do not hesitate to get in touch.

Sincerely,
Dr Robert Barton
<mailto:proceedingsb@royalsociety.org>

Associate Editor
Board Member
Comments to Author:

The revised manuscript has been reviewed again by the original reviewer 2, who is now happy that their original concerns have been addressed. The paper will make an important contribution to the literature.

Reviewer(s)' Comments to Author:

Referee: 2

Comments to the Author(s).
n/a

Sincerely,
Proceedings B
<mailto:proceedingsb@royalsociety.org>

Decision letter (RSPB-2020-1148.R1)

27-May-2020

Dear Ms Kamiloglu

I am pleased to inform you that your manuscript entitled "Human Listeners' Perception of Behavioural Context and Core Affect Dimensions in Chimpanzee Vocalisations" has been accepted for publication in Proceedings B.

Open Access

Paper charges

Sincerely,

Editor, Proceedings B
<mailto:proceedingsb@royalsociety.org>

Appendix A

Dear Professor Robert Barton,

We thank you and the referees for the useful comments on our manuscript RSPB-2020-0465 entitled "Human Listeners' Perception of Behavioural Context and Core Affect Dimensions in Chimpanzee Vocalisations" and the opportunity to resubmit a revised manuscript for consideration. We greatly appreciate the thoughtful feedback, which has helped us to improve our manuscript. We hope that you will find the revised manuscript suitable for publication in Proceedings B.

The changes made in response to each point raised by the referees are detailed in the point-point response below.

Referees' comments:

Referee #1: This is an interesting study that looked at humans' ability to categorize level of arousal, valence, and behavioural context from nonhuman primate vocalizations. Participants were good at the arousal and valence categorizations, but poorer at context categorizations.

I have no concerns, but I do wonder whether the authors administered any questionnaires to participants to determine their level of experience with nonhuman animals' vocalizations.

Thank you for this comment. We agree that participants' prior experience with vocalisations of nonhuman animals, especially chimpanzees, could influence their recognition accuracy. To ensure that the listeners had minimal prior exposure to chimpanzee vocalisations, we recruited participants who had no experience working with or studying chimpanzees; the recruitment text included the phrase "no experience working with or studying chimpanzees". Additionally, at the end of Experiment 1, we asked participants ($N = 300$) to report their familiarity with each behavioural context (How familiar are you with the chimpanzees in the context of X (e.g., discovering a large food source) from zoo settings or media?), and a representative vocalisation from each context (How familiar are you with this chimpanzee vocalization from zoo settings or media?) on a 5-point scale ('1 = not at all', '2 = slightly', '3 = moderately', '4 = very', '5 = extremely'). In the revised manuscript, we now report this measure (p.8/31, line 168):

"Finally, we participants reported their familiarity with both each behavioural context (How familiar are you with the chimpanzees in the context of X (e.g., discovering a large food source) from zoo settings or media?), and a representative vocalisation from each context (How familiar are you with this chimpanzee vocalization from zoo settings or media?) on a 5-point scale (1 = not at all, 5 = extremely)."

The results show that, participants rated behavioural contexts as less than "Slightly familiar" on average, and representative vocalisations as less than "Moderately familiar". We report this in the revised manuscript (p.10/31, line 232):

"On average, on the 1-5 likert scale where 1 = not at all familiar, participants rated both behavioural contexts ($M = 1.86$, $SD = 0.89$) and representative vocalisations ($M = 2.14$ $SD = 0.98$) as unfamiliar."

These results indicate that the listeners were not familiar with the chimpanzee vocalisations prior to our experiment. Moreover, they were not familiar with the behavioural contexts, suggesting that the behavioural context categorisation task used in Experiment 1 was likely to have been challenging for the listeners, as we also suggest.

Referee #2

Overall, I found this manuscript to be a generally thorough and interesting examination of cross-species vocal perception. It contains very detailed acoustic analyses and also two main experiments to determine human perception of chimp calls, based on behavioural context in which the calls were uttered, as well as arousal and valence (two key components linked to emotions). The results mostly appear robust and convincing, except see my comments on Exp 1 below.

I suggest the following changes:

Please insert line numbers in order to make the lives of reviewers and editors a little easier.

Thank you for pointing this out. Line numbers have been added in the revised manuscript.

Key words should be in alphabetical order.

Thank you, this has been corrected.

The order of results should be changed. A full understanding of the Experiments 1 and 2 is based on section 3 of the results - Acoustic Analysis. Reading Experiment 1 at first, I thought the classification of arousal in calls was based on subjective assessments of one of the authors (See: Text S1: Recording of chimpanzee vocalisations). Table 1. I assume the classification of calls according to arousal and valence is based on the results of the Acoustic Analysis section? However, this is not clear. For example, it is possible that, at least subjectively and without acoustic data, Tantrum screams might be considered High instead of Medium arousal, and Whimpers might be considered Low arousal instead of Medium.

Thank you for pointing us to the fact that the arousal and valence classification was unclear. The classifications of arousal (and valence) levels were determined by one of the authors, K.E.S., who is an expert on chimpanzee vocal communication. K.E.S. has over 15 years of experience studying chimpanzee communication, including long term behavioural research on both wild and captivity populations of chimpanzees. In previous research, it is common to use expert classifications of levels of arousal (e.g., Kelly et al., 2017) as well as valence (e.g., Belin et al., 2008; Braby, Shapira & Simmons, 2001; Maigrot, Hillmann, & Briefer, 2018; Scheumann, Hastin, Kotz, & Zimmenmann, 2014) from animal vocalisations. To clarify our approach, we have added the following part to the revised manuscript (p.7/31, line 148):

“The behavioural contexts were recorded by author K.E.S. in real time, alongside the sound recordings of vocalisations, and K.E.S., an expert in chimpanzee vocal communication, provided classifications of the arousal level (high, medium, low) and valence (positive, negative) of each call type (see Table 1).”

K.E.S used her knowledge, accrued from years of direct observation of chimpanzee vocal behaviour, of the vocaliser’s typical behaviour, the response of other individuals, and the context to provide the classifications of arousal and valence of each call type (Table 1).

In Experiments 1 and 2, we test whether naive participants can infer arousal levels and valence from chimpanzee vocalisations. Their results are consistent with the expert classifications of arousal and valence. In section 3, we conduct a classification analysis based on acoustic features (p.18/31, line 399) in order to test whether the expert’s and lay listeners’

classifications map onto differential acoustic configurations. We found that the acoustic features of the chimpanzee vocalisations varied systematically along the arousal and valence levels as determined by the expert and lay judgments.

We agree that physiological measures (e.g., heart rate) could help inform arousal and valence classifications. However, the field currently lacks adequate methods for dynamically capturing physiological arousal measures in free-moving, naturally behaving individuals, that would be necessary to connect physiological measures with specific vocal production events.

Braby, R. J., Shapira, A., & Simmons, R. E. (2001). Successful conservation measures and new breeding records for Damara Terns *Sterna balaenarum* in Namibia. *Marine Ornithology*, 29, 81-84.

Belin, P., Fecteau, S., Charest, I., Nicastro, N., Hauser, M. D., & Armony, J. L. (2008). Human cerebral response to animal affective vocalizations. *Proceedings of the Royal Society B: Biological Sciences*, 275(1634), 473–481.
<https://doi.org/10.1098/rspb.2007.1460>

Kelly, T., Reby, D., Levréro, F., Keenan, S., Gustafsson, E., Koutseff, A., & Mathevon, N. (2017). Adult human perception of distress in the cries of Bonobo, chimpanzee, and human infants. *Biological Journal of the Linnean Society*, 120(4), 919–930.
<https://doi.org/10.1093/biolinnean/blw016>

Maigrot, A. L., Hillmann, E., & Briefer, E. F. (2018). Encoding of emotional valence in wild boar (*Sus scrofa*) calls. *Animals*, 8(6), 85. <https://doi.org/10.3390/ani8060085>

Scheumann, M., Hasting, A. S., Kotz, S. A., & Zimmermann, E. (2014). The voice of emotion across species: How do human listeners recognize animals' affective states? *PLoS ONE*, 9(3), 1–10. <https://doi.org/10.1371/journal.pone.0091192>

Please provide examples of the calls types as Supplemental files. These should be available at the review stage.

We previously provided example vocalisations on a website, which we included a link to in the manuscript (<https://emotionwaves.github.io/chimp/>, p. 7/32). In addition, in the revised manuscript we now include the sound files as Supplementary Materials Audio 1S and refer to this source in p.7/31, line 154.

Please insert the sample size for chimps used in the study and not just the number of vocalisations ($n = 155$). The chimp sample size used for sourcing the call examples also needs to be more prominent in both the Methods (e.g. page 7/32) and Results of the manuscript – currently it is quite difficult to locate.

Thank you for this suggestion to include the chimpanzee sample size ($n = 66$). We have added this information to the Abstract (p. 2/31. line 29), Materials and Procedure section of the Experiment 1 (p. 6/31, line 146), and Method section of the Acoustic Analysis section (p. 16/31, line 338).

Page 8/32. Experiment 1. “Participants listened to the 155 chimpanzee vocalisations”. I have reservations about how informative this sort of setup could be (with 10 behavioural contexts, three arousal levels). How long (average and SD) did it take human participants to work their way through 155 chimp calls? Would you expect the same level of accuracy and focus in the

human subjects at call 10 or 14, versus call 145 or 150? There is no analysis reported that checks whether the human subjects were better at classifying the first 30 calls versus the last 30, for example.

Thank you for making this very good point. We agree that the task used in Experiment 1 was challenging for participants: It required attentively making three judgements for 155 chimpanzee vocalisations. The experiment took participants 27.43 minutes to complete on average ($SD = 9.75$). We now note this in the revised manuscript (p. 6/31, line 141):

“The average duration of the main experiment was 27.43 minutes ($SD = 9.75$), and participation was compensated with monetary reward or course credit.”

Following the referee’s comment, we compared accuracy in the first 30 trials against accuracy in the last 30 trials. Using pairwise comparisons, we tested whether performance accuracy (in behavioural context categorisation task, as well as judgements of arousal and valence) would differ between the first and last 30 trials of the task. The results show that performance accuracy did not change significantly for context categorisation and valence judgements. However, for judgements of arousal level, participants performed better in the first, as compared to the last, 30 trials ($z = 2.552, p = 0.011$).

The results thus show a decline in performance accuracy for arousal judgments from the early to late trials, while there was no such difference in accuracy for the other types of judgements. However, participants’ performance on the arousal judgement task was high also in the last 30 trials ($M = 44.92, SD = 0.11$) when compared to first 30 trials ($M = 46.74, SD = 0.10$). Nevertheless, to minimise listener fatigue, we used a less taxing task in Experiment 2.

The analysis reporting the results of this comparison is now provided in Supplementary Materials Table 1S, and we point the reader to it in the revised manuscript (p. 10/31, line 234):

“Because of the large number of stimuli and judgements, we checked for evidence of fatigue by comparing the accuracy in early (the first 30) and late (the last 30) trials. Pairwise comparisons showed that participants’ performance on the arousal judgement task was high in both the early ($M = 46.74, SD = 0.10$) and late trials ($M = 44.92, SD = 0.11$), although participants performed better in the early trials ($z = 2.552, p = 0.011$). No difference in accuracy was found for early and late judgments of context categorisation and valence (see Supplementary Materials Table 1S for details). It is therefore unlikely that participants’ judgement performance was affected by possible fatigue.”

Page 19/32. MNR?

Thank you for pointing out this error. We now corrected it by changing MNR to MLR, which refers to Multinomial Logistic Regression in p.18/31, lines 400 and 406 in the revised manuscript.

The formatting of references contains many inconsistencies, which are not in the journal style.

Thank you for this comment. All references have been edited to journal style in the revised manuscript.

Sincerely,
Roza G. Kamiloglu, on behalf of the authors